# SIMPLE: A GRADIENT ESTIMATOR FOR $k$-SUBSET SAMPLING

**Kareem Ahmed**[*1]    **Zhe Zeng**[*1]    **Mathias Niepert**[2]    **Guy Van den Broeck**[1]

[1]Dept. of Computer Science, UCLA    [2]Dept. of Computer Science, Stuttgart University

`{ahmedk,zhezeng,guyvdb}@cs.ucla.edu`
`mathias.niepert@simtech.uni-stuttgart.de`

## ABSTRACT

$k$-subset sampling is ubiquitous in machine learning, enabling regularization and interpretability through sparsity. The challenge lies in rendering $k$-subset sampling amenable to end-to-end learning. This has typically involved *relaxing* the *reparameterized* samples to allow for backpropagation, with the risk of introducing high bias and high variance. In this work, we fall back to *discrete* $k$-subset sampling on the forward pass. This is coupled with using the gradient with respect to the *exact* marginals, computed *efficiently*, as a proxy for the true gradient. We show that our gradient estimator, SIMPLE, exhibits lower bias and variance compared to state-of-the-art estimators, including the straight-through Gumbel estimator when $k = 1$. Empirical results show improved performance on learning to explain and sparse linear regression. We provide an algorithm for computing the *exact* ELBO for the $k$-subset distribution, obtaining *significantly* lower loss compared to SOTA.

## 1 INTRODUCTION

$k$-subset sampling, sampling a subset of size $k$ of $n$ variables, is omnipresent in machine learning. It lies at the core of many fundamental problems that rely upon learning sparse features representations of input data, including stochastic high-dimensional data visualization (van der Maaten, 2009), parametric $k$-nearest neighbors (Grover et al., 2018), learning to explain (Chen et al., 2018), discrete variational auto-encoders (Rolfe, 2017), and sparse regression, to name a few. All such tasks involve optimizing an expectation of an objective function with respect to a latent *discrete* distribution parameterized by a neural network, which are often *assumed* intractable. Score-function estimators offer a cloyingly simple solution: rewrite the gradient of the expectation as an expectation of the gradient, which can subsequently be estimated using a finite number of samples offering an unbiased estimate of the gradient. Simple as it is, score-function estimators suffer from very high variance which can interfere with training. This provided the impetus for other, low-variance, gradient estimators, chief among them are those based on the reparameterization trick, which allows for biased, but low-variance gradient estimates. The reparameterization trick, however, does not allow for a direct application to discrete distributions thereby prompting continuous relaxations, e.g. Gumbel-softmax (Jang et al., 2017; Maddison et al., 2017), that allow for reparameterized gradients w.r.t the parameters of a *categorical* distribution. Reparameterizable subset sampling (Xie & Ermon, 2019) generalizes the Gumbel-softmax trick to $k$-subsets which while rendering $k$-subset sampling amenable to backpropagation at the cost of introducing bias in the learning by using relaxed samples.

In this paper, we set out with the goal of avoiding all such relaxations. Instead, we fall back to *discrete* sampling on the forward pass. On the backward pass, we reparameterize the gradient of the loss function with respect to the samples as a function of the *exact* marginals of the $k$-subset distribution. Computing the exact conditional marginals is, in general, intractable (Roth, 1996). We give an *efficient* algorithm for computing the $k$-subset probability, and show that the conditional marginals correspond to partial derivatives, and are therefore tractable for the $k$-subset distribution. We show that our proposed gradient estimator for the $k$-subset distribution, coined SIMPLE, is reminiscent of the straight-through (ST) Gumbel estimator when $k = 1$, with the gradients taken with respect to the unperturbed marginals. We empirically demonstrate that SIMPLE exhibits lower bias *and* variance compared to other known gradient estimators, including the ST Gumbel estimator in the case $k = 1$.

---

[*]These authors contributed equally to this work.

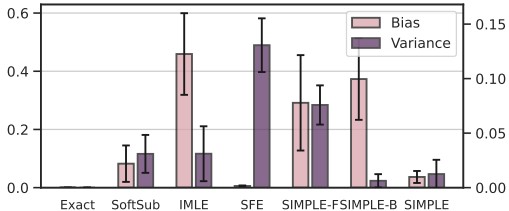 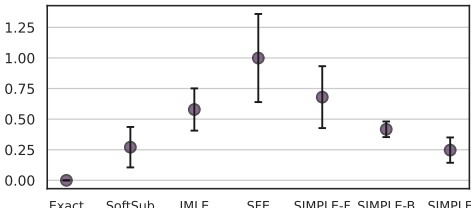

Figure 1: A comparison of the bias and variance of the gradient estimators (left) and the average and standard deviation of the cosine distance of a single-sample gradient estimate to the exact gradient. We used the cosine distance, defined as $(1-$ cosine similarity), in place of the euclidean distance as we only care about the direction of the gradient, not magnitude. The bias, variance and error were estimated using a sample of size 10,000. The details of this experiment are provided in Section 5.1.

We include an experiment on the task of learning to explain (L2X) using the BEERADVOCATE dataset (McAuley et al., 2012), where the goal is to select the subset of words that best explains the model's classification of a user's review. We also include an experiment on the task of stochastic sparse linear regression, where the goal is to learn the best sparse model, and show that we are able to recover the Kuramoto–Sivashinsky equation. Finally, we develop an efficient computation for the calculation of the exact variational evidence lower bound (ELBO) for the $k$-subset distribution, which when used in conjunction with SIMPLE leads to state-of-the-art discrete sparse VAE learning.

**Contributions.** In summary, we propose replacing *relaxed* sampling on the forward pass with *discrete* sampling. On the backward pass, we use the gradient with respect to the exact conditional marginals as a proxy for the true gradient, giving an algorithm for computing them efficiently. We empirically demonstrate that discrete samples on the forward pass, coupled with exact conditional marginals on the backward pass leads to a new gradient estimator, SIMPLE, with lower bias and variance compared to other known gradient estimators. We also provide an efficient computation of the exact ELBO for the $k$-subset distribution, leading to state-of-the-art discrete sparse VAE learning.

## 2 PROBLEM STATEMENT AND MOTIVATION

We consider models described by the equations
$$\boldsymbol{\theta} = h_{\boldsymbol{v}}(\boldsymbol{x}), \qquad \boldsymbol{z} \sim p_{\boldsymbol{\theta}}(\boldsymbol{z} \mid \textstyle\sum_i z_i = k), \qquad \hat{\boldsymbol{y}} = f_{\boldsymbol{u}}(\boldsymbol{z}, \boldsymbol{x}), \tag{1}$$
where $\boldsymbol{x} \in \mathcal{X}$ and $\hat{\boldsymbol{y}} \in \mathcal{Y}$ denote feature inputs and target outputs, respectively, $h_{\boldsymbol{v}} : \mathcal{X} \to \Theta$ and $f_{\boldsymbol{u}} : \mathcal{Z} \times \mathcal{X} \to \mathcal{Y}$ are smooth, parameterized maps and $\boldsymbol{\theta}$ are logits inducing a distribution over the latent binary vector $\boldsymbol{z}$. The induced distribution $p_{\boldsymbol{\theta}}(\boldsymbol{z})$ is defined as
$$p_{\boldsymbol{\theta}}(\boldsymbol{z}) = \prod_{i=1}^{n} p_{\theta_i}(z_i), \text{ with } p_{\theta_i}(z_i = 1) = \text{sigmoid}(\theta_i) \text{ and } p_{\theta_i}(z_i = 0) = 1 - \text{sigmoid}(\theta_i). \tag{2}$$
The goal of our stochastic latent layer is *not* to simply sample from $p_{\boldsymbol{\theta}}(\boldsymbol{z})$, which would yield samples with a Hamming weight between $0$ and $n$ (i.e., with an arbitrary number of ones). Instead, we are interested in sampling from the distribution restricted to samples with a Hamming weight of $k$, for any given $k$. That is, we are interested in sampling from the conditional distribution $p_{\boldsymbol{\theta}}(\boldsymbol{z} \mid \sum_i z_i = k)$.

Conditioning the distribution $p_{\boldsymbol{\theta}}(\boldsymbol{z})$ on this $k$-*subset constraint* introduces intricate dependencies between each of the $z_i$'s. The probability of sampling any given $k$-subset vector $\boldsymbol{z}$, therefore, becomes
$$p_{\boldsymbol{\theta}}(\boldsymbol{z} \mid \textstyle\sum_i z_i = k) = p_{\boldsymbol{\theta}}(\boldsymbol{z}) / p_{\boldsymbol{\theta}}(\sum_i z_i = k) \cdot [\![\sum_i z_i = k]\!]$$
where $[\![\cdot]\!]$ denotes the indicator function. In other words, the probability of sampling each $k$-subset is re-normalized by $p_{\boldsymbol{\theta}}(\sum_i z_i = k)$ – the probability of sampling exactly $k$ items from the *unconstrained* distribution induced by encoder $h_{\boldsymbol{v}}$. The quantity $p_{\boldsymbol{\theta}}(\sum_i z_i = k) = \sum_{\boldsymbol{z}} p_{\boldsymbol{\theta}}(\boldsymbol{z}) \cdot [\![\sum_i z_i = k]\!]$ appears to be intractable. We show that not to be the case, providing a tractable algorithm for computing it.

Given a set of samples $\mathcal{D}$, we are concerned with learning the parameters $\boldsymbol{\omega} = (\boldsymbol{v}, \boldsymbol{u})$ of the architecture in (1) through minimizing the training error $L$, which is the expected loss:
$$L(\boldsymbol{x}, \boldsymbol{y}; \boldsymbol{\omega}) = \mathbb{E}_{\boldsymbol{z} \sim p_{\boldsymbol{\theta}}(\boldsymbol{z} \mid \sum_i z_i = k)}[\ell(f_{\boldsymbol{u}}(\boldsymbol{z}, \boldsymbol{x}), \boldsymbol{y})] \qquad \textit{with } \boldsymbol{\theta} = h_{\boldsymbol{v}}(\boldsymbol{x}), \tag{3}$$

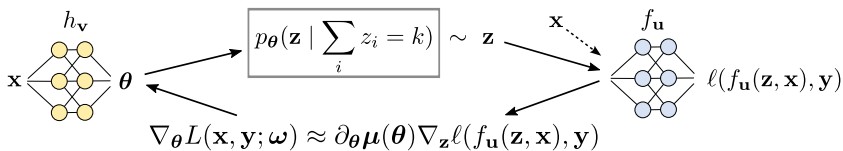

Figure 2: The problem setting considered in our paper. On the forward pass, a neural network $h_v$ outputs $\boldsymbol{\theta}$ parameterizing a *discrete* distribution over subsets of size $k$ of $n$ items, i.e., the $k$-subset distribution. We sample exactly, and efficiently, from this distribution, and feed the samples to a downstream neural network. On the backward pass, we approximate the true gradient by the product of the derivative of marginals and the gradient of the sample-wise loss.

where $\ell : \mathcal{Y} \times \mathcal{Y} \to \mathbb{R}^+$ is a point-wise loss function. This formulation, illustrated in Figure 2, is general and subsumes many settings. Different choices of mappings $h_v$ and $f_u$, and sample-wise loss $\ell$ define various tasks. Table 1 presents some example settings used in our experimental evaluation.

| TASK | MAP $h_v$ | MAP $f_u$ | LOSS $\ell$ |
|---|---|---|---|
| Discrete VAE (Sec. 5.2) | Encoder | Decoder | ELBO |
| Learn To Explain (Sec. 5.3) | Embedding | Regression | RMSE |
| Sparse Regression (Sec. 5.4) | Identity | Linear Regression | RMSE |

Table 1: Architectures of the three experiment settings.

Learning then requires computing the gradient of $L$ w.r.t. $\boldsymbol{\omega} = (\boldsymbol{v}, \boldsymbol{u})$. The gradient of $L$ w.r.t. $\boldsymbol{u}$ is

$$\nabla_{\boldsymbol{u}} L(\boldsymbol{x}, \boldsymbol{y}; \boldsymbol{\omega}) = \mathbb{E}_{\boldsymbol{z} \sim p_{\boldsymbol{\theta}}(\boldsymbol{z} | \sum_i z_i = k)} [\partial_{\boldsymbol{u}} f_{\boldsymbol{u}}(\boldsymbol{z}, \boldsymbol{x})^\top \nabla_{\hat{\boldsymbol{y}}} \ell(\hat{\boldsymbol{y}}, \boldsymbol{y})], \tag{4}$$

where $\hat{y} = f_{\boldsymbol{u}}(\boldsymbol{z}, \boldsymbol{x})$ is the decoding of a latent sample $\boldsymbol{z}$. Furthermore, the gradient of $L$ w.r.t. $\boldsymbol{v}$ is

$$\nabla_{\boldsymbol{v}} L(\boldsymbol{x}, \boldsymbol{y}; \boldsymbol{\omega}) = \partial_{\boldsymbol{v}} h_{\boldsymbol{v}}(\boldsymbol{x})^\top \nabla_{\boldsymbol{\theta}} L(\boldsymbol{x}, \boldsymbol{y}; \boldsymbol{\omega}), \tag{5}$$

where $\nabla_{\boldsymbol{\theta}} L(\boldsymbol{x}, \boldsymbol{y}; \boldsymbol{\omega}) := \nabla_{\boldsymbol{\theta}} \mathbb{E}_{\boldsymbol{z} \sim p_{\boldsymbol{\theta}}(\boldsymbol{z} | \sum_i z_i = k)}[\ell(f_{\boldsymbol{u}}(\boldsymbol{z}, \boldsymbol{x}), \hat{\boldsymbol{y}})]$, the loss' gradient w.r.t. the encoder.

One challenge lies in computing the expectation in (3) and (4), which has no known closed-form solution. This necessitates a Monte-Carlo estimate through sampling from $p_{\boldsymbol{\theta}}(\boldsymbol{z} | \sum_i z_i = k)$.

A second, and perhaps more substantial hurdle lies in computing $\nabla_{\boldsymbol{\theta}} L(\boldsymbol{x}, \boldsymbol{y}; \boldsymbol{\omega})$ in (5) due to the non-differentiable nature of discrete sampling. One could rewrite $\nabla_{\boldsymbol{\theta}} L(\boldsymbol{x}, \boldsymbol{y}; \boldsymbol{\omega})$ as

$$\nabla_{\boldsymbol{\theta}} L(\boldsymbol{x}, \boldsymbol{y}; \boldsymbol{\omega}) = \mathbb{E}_{\boldsymbol{z} \sim p_{\boldsymbol{\theta}}(\boldsymbol{z} | \sum_i z_i = k)}[\ell(f_{\boldsymbol{u}}(\boldsymbol{z}, \boldsymbol{x}), \boldsymbol{y}) \nabla_{\boldsymbol{\theta}} \log p_{\boldsymbol{\theta}}(\boldsymbol{z} | \sum_i z_i = k)]$$

which is known as the REINFORCE estimator (Williams, 1992), or the score function estimator (SFE). It is typically avoided due to its notoriously high variance, despite its apparent simplicity. Instead, typical approaches (Xie & Ermon, 2019; Plötz & Roth, 2018) reparameterize the samples as a deterministic transformation of the parameters, and some independent standard Gumbel noise, and relaxing the deterministic transformation, the top-$k$ function in this case, to allow for backpropagation.

# 3 SIMPLE: SUBSET IMPLICIT LIKELIHOOD ESTIMATION

Our goal is to build a gradient estimator for $\nabla_{\boldsymbol{\theta}} L(\boldsymbol{x}, \boldsymbol{y}; \boldsymbol{\omega})$. We start by envisioning a hypothetical sampling-free architecture, where the downstream neural network $f_{\boldsymbol{u}}$ is a function of the marginals, $\boldsymbol{\mu} := \mu(\boldsymbol{\theta}) := \{p_{\boldsymbol{\theta}}(z_j | \sum_i z_i = k)\}_{j=1}^n$, instead of a discrete sample $\boldsymbol{z}$, resulting in a loss $L_m$ s.t.

$$\nabla_{\boldsymbol{\theta}} L_m(\boldsymbol{x}, \boldsymbol{y}; \boldsymbol{\omega}) = \partial_{\boldsymbol{\theta}} \mu(\boldsymbol{\theta})^\top \nabla_{\boldsymbol{\mu}} \ell_m(f_{\boldsymbol{u}}(\boldsymbol{\mu}, \boldsymbol{x}), \boldsymbol{y}). \tag{6}$$

When the marginals $\mu(\boldsymbol{\theta})$ can be efficiently computed and differentiated, such a hypothetical pipeline can be trained end-to-end. Furthermore, Domke (2010) observed that, for an arbitrary loss function $\ell_m$ defined on the marginals, the Jacobian of the marginals w.r.t. the logits is symmetric, i.e.

$$\nabla_{\boldsymbol{\theta}} L_m(\boldsymbol{x}, \boldsymbol{y}; \boldsymbol{\omega}) = \partial_{\boldsymbol{\theta}} \mu(\boldsymbol{\theta})^\top \nabla_{\boldsymbol{\mu}} \ell_m(f_{\boldsymbol{u}}(\boldsymbol{\mu}, \boldsymbol{x}), \boldsymbol{y}) = \partial_{\boldsymbol{\theta}} \mu(\boldsymbol{\theta}) \nabla_{\boldsymbol{\mu}} \ell_m(f_{\boldsymbol{u}}(\boldsymbol{\mu}, \boldsymbol{x}), \boldsymbol{y}). \tag{7}$$

Consequently, computing the gradient of the loss w.r.t. the logits, $\nabla_{\boldsymbol{\theta}} L_m(\boldsymbol{x}, \boldsymbol{y}; \boldsymbol{\omega})$, reduces to computing the *directional derivative*, or the Jacobian-vector product, of the marginals w.r.t. the logits in the direction of the gradient of the loss. This offers an alluring opportunity: the conditional

---

**Algorithm 1** PrExactlyk($\boldsymbol{\theta}, n, k$)

---

**Input**: The logits $\boldsymbol{\theta}$ of the distribution, the number of variables $n$, and the subset size $k$
**Output**: $p_{\boldsymbol{\theta}}(\sum_i z_i = k)$
   // $a[i,j] = p_{\boldsymbol{\theta}}(\sum_{m=1}^{i} z_m = j)$ for all $i, j$
   initialize $a$ to be 0 everywhere
   $a[0,0] = 1$ // $p_{\boldsymbol{\theta}}(\sum_{m=1}^{0} z_m = 0) = 1$
   **for** $i = 1$ **to** $n$ **do**
      **for** $j = 0$ **to** $k$ **do**
         // cf. constructive proof of Prop. 1
         $a[i,j] = a[i-1,j] \cdot p_{\theta_i}(z_i = 0)$
              $+ \, a[i-1,j-1] \cdot p_{\theta_i}(z_i = 1)$
   **return** $a[n,k]$

---

**Algorithm 2** Sample($\boldsymbol{\theta}, n, k$)

---

**Input**: The logits $\boldsymbol{\theta}$ of the distribution, the number of variables $n$, and the subset size $k$
**Output**: $\boldsymbol{z} = (z_1, \ldots, z_n) \sim p_{\boldsymbol{\theta}}(\boldsymbol{z} \mid \sum_i z_i = k)$
   sample = [ ], $j = k$
   **for** $i = n$ **to** $1$ **do**
      // cf. proof of Prop. 2
      $p = a[i-1, j-1]$
      $z_i \sim \text{Bernoulli}(p \cdot p_{\theta_i}(z_i = 1)/a[i,j])$
      // Pick next state based on value of sample
      **if** $z_i = 1$ **then** $j = j - 1$
      sample.append($z_i$)
   **return** sample

---

marginals characterize the probability of each $z_i$ in the sample, and could be thought of as a differentiable proxy for the samples. Specifically, by reparameterizing $\boldsymbol{z}$ as a function of the conditional marginal $\boldsymbol{\mu}$ under approximation $\partial_{\boldsymbol{\mu}} \boldsymbol{z} \approx \mathbf{I}$ as proposed by Niepert et al. (2021), and using the straight-through estimator for the gradient of the sample w.r.t. the marginals on the backward pass, we approximate our true $\nabla_{\boldsymbol{\theta}} L(\boldsymbol{x}, \boldsymbol{y}; \boldsymbol{\omega})$ as

$$\nabla_{\boldsymbol{\theta}} L(\boldsymbol{x}, \boldsymbol{y}; \boldsymbol{\omega}) \approx \partial_{\boldsymbol{\theta}} \mu(\boldsymbol{\theta}) \nabla_{\boldsymbol{z}} L(\boldsymbol{x}, \boldsymbol{y}; \boldsymbol{\omega}), \qquad (8)$$

where the directional derivative of the marginals can be taken along *any downstream gradient*, rendering the whole pipeline end-to-end learnable, even in the presence of non-differentiable sampling.

Now, estimating the gradient of the loss w.r.t. the parameters can be thought of as decomposing into two sub-problems: **(P1)** Computing the derivatives of conditional marginals $\partial_{\boldsymbol{\theta}} \mu(\boldsymbol{\theta})$, which requires the computation of the conditional marginals, and **(P2)** Computing the gradient of the loss w.r.t. the samples $\nabla_{\boldsymbol{z}} L(\boldsymbol{x}, \boldsymbol{y}; \boldsymbol{\omega})$ using sample-wise loss, which requires drawing exact samples. These two problems are complicated by conditioning on the $k$-subset constraint, which introduces intricate dependencies to the distribution, and is infeasible to solve naively, e.g. by enumeration. We will show simple, efficient, and exact solutions to each problem, at the heart of which is the insight that we need not care about the variables' order, only their sum, introducing symmetries that simplify the problem.

### 3.1 DERIVATIVES OF CONDITIONAL MARGINALS

In many probabilistic models, marginal inference is #P-hard (Roth, 1996; Zeng et al., 2020). However, we observe that it is not the case for the $k$-subset distribution. We notice that the conditional marginals correspond to the partial derivatives of the log-probability of the $k$-subset constraint. To see this, note that the derivative of a multi-linear function with respect to a single variable retains all the terms referencing that variable, and drops all other terms; this corresponds exactly to the unnormalized conditional marginals. By taking the derivative of the log-probability, this introduces the $k$-subset probability in the denominator, leading to the *conditional* marginals. Intuitively, the rate of change of the $k$-subset probability w.r.t. a variable only depends on that variable through its length-$k$ subsets.

**Theorem 1.** *Let $p_{\boldsymbol{\theta}}(\sum_j z_j = k)$ be the probability of exactly-$k$ of the unconstrained distribution parameterized by logits $\boldsymbol{\theta}$. For every variable $Z_i$, its conditional marginal is*

$$p_{\boldsymbol{\theta}}\Big(z_i \mid \sum_j z_j = k\Big) = \frac{\partial}{\partial \theta_i} \log p_{\boldsymbol{\theta}}(\sum_j z_j = k). \qquad (9)$$

We refer the reader to the appendix for a detailed proof of the above theorem. To establish the tractability of the above computation of the conditional marginals, we need to show that the probability of the exactly-$k$ constraint $p_{\boldsymbol{\theta}}(\sum_i z_i = k)$ can be obtained tractably, which we demonstrate next.

**Proposition 1.** *The probability $p_{\boldsymbol{\theta}}(\sum_i z_i = k)$ of sampling exactly $k$ items from the unconstrained distribution $p_{\boldsymbol{\theta}}(\boldsymbol{z})$ over $n$ items as in Equation 2 can be computed exactly in time $\mathcal{O}(nk)$.*

*Proof.* Our proof is constructive. As a base case, consider the probability of sampling $k = -1$ out of $n = 0$ items. We can see that the probability of such an event is 0. As a second base case, consider

the probability of sampling $k = 0$ out of $n = 0$ items. We can see that the probably of such an event is 1. Now assume that we are given the probability $p_{\boldsymbol{\theta}} \left( \sum_i^{n-1} z_i = k' \right)$, for $k' = 0, \ldots, k$, and we are interested in computing $p_{\boldsymbol{\theta}} \left( \sum_i^n z_i = k \right)$. By the partition theorem, we can see that

$$p_{\boldsymbol{\theta}} \left( \sum_i^n z_i = k \right) = p_{\boldsymbol{\theta}} \left( \sum_i^{n-1} z_i = k \right) \cdot p_{\theta_n}(z_n = 0) + p_{\boldsymbol{\theta}} \left( \sum_i^{n-1} z_i = k - 1 \right) \cdot p_{\theta_n}(z_n = 1)$$

as events $\sum_i^{n-1} z_i = k$ and $\sum_i^{n-1} z_i = k - 1$ are disjoint and, for any $k$, partition the sample space. Intuitively, for any $k$ and $n$, we can sample $k$ out of $n$ items by choosing $k$ of $n - 1$ items, and not the $n$-th item, or choosing $k - 1$ of $n - 1$ items, and the $n$-th item. The above process gives rise to Algorithm 1, which returns $p_{\boldsymbol{\theta}} \left( \sum_i z_i = k \right)$ in time $\mathcal{O}(nk)$. $\qquad\square$

By the construction described above, we obtain a closed-form $p_{\boldsymbol{\theta}}(\sum_i^n z_i = k)$, which allows us to compute conditional marginals $p_{\boldsymbol{\theta}}(z_i \mid \sum_j z_j = k)$ by Theorem 1 via auto-differentiation. This further allows the computation of the derivatives of conditional marginals $\partial_{\boldsymbol{\theta}} \mu(\boldsymbol{\theta})_i = \partial_{\boldsymbol{\theta}} \, p_{\boldsymbol{\theta}}(z_i \mid \sum_j z_j = k)$ to be amenable to auto-differentiation, solving problem (**P1**) exactly and efficiently.

## 3.2 Gradients of Loss w.r.t. Samples

As alluded to in Section 3, we approximate $\nabla_{\boldsymbol{\theta}} L(\boldsymbol{x}, \boldsymbol{y}; \boldsymbol{\omega})$ by the directional derivative of the marginals along the gradient of the loss w.r.t. discrete samples $\boldsymbol{z}$, $\nabla_{\boldsymbol{z}} L(\boldsymbol{x}, \boldsymbol{y}; \boldsymbol{\omega})$, where $\boldsymbol{z}$ is drawn from the $k$-subset distribution $p_{\boldsymbol{\theta}}(\boldsymbol{z} \mid \sum_i z_i = k)$. What remains is to estimate the value of the loss, necessitating faithful sampling from the $k$-subset distribution, which might initially appear daunting.

**Exact $k$-subset Sampling** Next we show how to sample exactly from the $k$-subset distribution $p_{\boldsymbol{\theta}}(\boldsymbol{z} \mid \sum_i z_i = k)$. We start by sampling the variables in reverse order, that is, we sample $z_n$ through $z_1$. The main intuition being that, having sampled $(z_n, z_{n-1}, \cdots, z_{i+1})$ with a Hamming weight of $k - j$, we sample $Z_i$ with a probability of choosing $k - j$ of $n - 1$ variables *and* the $n$-th variable *given that* we choose $k - j + 1$ of $n$ variables. We formalize our intuition below.

**Proposition 2.** *Let* Sample *be defined as in Algorithm 2. Given $n$ random variables $Z_1, \cdots, Z_n$, a subset size $k$, and a $k$-subset distribution $p_{\boldsymbol{\theta}}(\boldsymbol{z} \mid \sum_i z_i = k)$ parameterized by log probabilities $\boldsymbol{\theta}$, Algorithm 2 draws exact samples from $p_{\boldsymbol{\theta}}(\boldsymbol{z} \mid \sum_i z_i = k)$ in time $\mathcal{O}(n)$.*

*Proof.* Assume that variables $Z_n, \cdots, Z_{i+1}$ are sampled and have their values to be $z_n, \cdots, z_{i+1}$ with $\sum_{m=i+1}^n z_m = k - j$. By Algorithm 2 we have that the probability with which to sample $Z_i$ is

$$p_{\texttt{Sample}}(z_i = 1 \mid z_n, \cdots, z_{i+1}) = \frac{p_{\boldsymbol{\theta}}(\sum_{m=i}^n z_m = k - j + 1 \mid \sum_m z_m = k) \, p_{\theta_i}(z_i = 1)}{p_{\boldsymbol{\theta}}(\sum_{m=i+1}^n z_m = k - j \mid \sum_m z_m = k)}$$

$$= \frac{p_{\boldsymbol{\theta}}(\sum_{m=i+1}^n z_m = k - j \mid z_i = 1, \sum_m z_m = k) \, p_{\theta_i}(z_i = 1)}{p_{\boldsymbol{\theta}}(\sum_{m=i+1}^n z_m = k - j \mid \sum_m z_m = k)}$$

$$= p_{\boldsymbol{\theta}}(z_i = 1 \mid \sum_{m=i+1}^n z_m = k - j, \sum_m z_m = k) \textit{ (by Bayes' theorem)}$$

It follows that samples drawn from Algorithm 2 are distributed according to $p_{\boldsymbol{\theta}}(\boldsymbol{z} \mid \sum_i z_i = k)$. $\quad\square$

## 4 Connection to Straight-Through Gumbel-Softmax

One might wonder if our gradient estimator reduces to the Straight-Through (ST) Gumbel-Softmax estimator, or relates to it in any way when $k = 1$. On the forward pass, the ST Gumbel Softmax estimator makes use of the Gumbel-Max trick (Maddison et al., 2014), which states that we can efficiently sample from a categorical distribution by perturbing each of the logits with standard Gumbel noise, and taking the MAP, or more formally $\boldsymbol{z} = \text{OneHot}(\arg\max_{i \in \{1, \ldots, k\}} \boldsymbol{\theta}_i + g_i) \sim p_{\boldsymbol{\theta}}$ where the $g_i$'s are i.i.d Gumbel$(0, 1)$ samples, and OneHot encodes the sample as a binary vector.

Since $\arg\max$ is non-differentiable, Gumbel-Softmax uses the *perturbed* relaxed samples, $\boldsymbol{y} = \text{Softmax}(\boldsymbol{\theta} + g_i)$ as a proxy for discrete samples $\boldsymbol{z}$ on the backward pass, using differentiable Softmax in place of the non-differentiable $\arg\max$, with the entire function returning $(\boldsymbol{z} - \boldsymbol{y}).\,\text{detach}() + \boldsymbol{y}$ where $\text{detach}$ ensures that the gradient flows only through the relaxed samples on the backward pass.

---

**Algorithm 3** The proposed algorithm for the $k$-subset distribution

---

**function** FORWARDPASS($\boldsymbol{\theta}$)
    *// $p_{\boldsymbol{\theta}}(\sum_{m=1}^{i} z_m = j)$ for all $i, j$*
    $a = \texttt{PrExactlyk}(\boldsymbol{\theta}, n, k)$
    *// Sample from $p_{\boldsymbol{\theta}}(\boldsymbol{z} \mid \sum_i z_i = k)$*
    $\boldsymbol{z} = \texttt{Sample}(\boldsymbol{\theta}, n, k)$
    **save** $a$ for the backward pass
    **return** $\boldsymbol{z}$

**function** BACKWARDPASS($\nabla_{\boldsymbol{z}} \ell(f_{\boldsymbol{u}}(\boldsymbol{z}, \boldsymbol{x}), \boldsymbol{y})$)
    **load** $\boldsymbol{\theta}$ from the forward pass
    *// derivatives of $p_{\boldsymbol{\theta}}(\boldsymbol{z} \mid \sum_i z_i = k)$*
    $\boldsymbol{\mu} = \nabla_{\boldsymbol{\theta}} \log a[n, k]$ *// by auto-diff*
    *// Return the directional derivative of the*
    *// marginals along the downstream gradients*
    **return** $\text{JVP}(\boldsymbol{\mu}, \nabla_{\boldsymbol{z}} \ell(f_{\boldsymbol{u}}(\boldsymbol{z}, \boldsymbol{x})))$

---

That is, just like SIMPLE, ST Gumbel-Softmax returns *exact*, *discrete* samples. However, whereas SIMPLE back-propagates through the exact marginals, ST Gumbel Soft-max backpropagates through the *perturbed* marginals that result from applying the Gumbel-max trick. As can be seen in Figure 3, such a minor difference means that, empirically, SIMPLE exhibits lower bias and variance compared to ST Gumbel Softmax while being exactly as efficient.

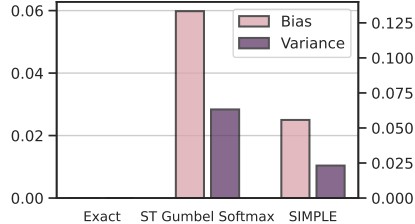

Figure 3: Bias and variance of SIMPLE and Gumbel Softmax over 10k samples

## 5 EXPERIMENTS

We conduct experiments on four different tasks: 1) A synthetic experiment designed to test the bias and variance, as well as the average deviation of SIMPLE compared to a variety of well-established estimators in the literature. 2) A discrete $k$-subset Variational Auto-Encoder (DVAE) setting, where the latent space models a probability distribution over $k$-subsets. We will show that we can compute the evidence lower bound (ELBO) exactly, and that, coupled with exact sampling and our SIMPLE gradient estimator, we attain a much lower loss compared to state of the art in sparse DVAEs. 3) The learning to explain (L2X) setting, where the aim is to select the $k$-subset of words that best describe the classifier's prediction, where we show an improved mean-squared error, as well as precision, across the board. 4) A novel, yet simple task, sparse linear regression, where, in a vein similar to L2X, we wish to select a $k$-subset of features that give rise to a linear regression model, avoiding overfitting the spurious features present in the data. Table 1 details the architecture with the objective functions. Our code will be made publicly available at github.com/UCLA-StarAI/SIMPLE.

### 5.1 SYNTHETIC EXPERIMENTS

We carried out a series of experiments with a 5-subset distribution, and a latent space of dimension 10. We set the loss to $L(\boldsymbol{\theta}) = \mathbb{E}_{\boldsymbol{z} \sim p_{\boldsymbol{\theta}}(\mathbf{z} \mid \sum_i z_i = k)}[\|\boldsymbol{z} - \mathbf{b}\|^2]$, where $\mathbf{b}$ is the groundtruth logits sampled from $\mathcal{N}(0, \mathbf{I})$. Such a distribution is tractable: we only have $\binom{10}{5} = 252$ $k$-subsets, which are easily enumerable and therefore, the exact gradient, the golden standard, can be computed in closed form.

In this experiment, we are interested in three metrics: bias, variance, and the average error of each gradient estimator, where the latter is measured by averaging the deviation of each single-sample gradient estimate from the exact gradient. We used the cosine distance, defined as $1-$ cosine similarity as the measure of deviation in our calculation of the metrics above, as we only care about direction.

We compare against four different baselines: *exact*, which denotes the exact gradient; *SoftSub* (Xie & Ermon, 2019), which uses an extension of the Gumbel-Softmax trick to sample *relaxed* $k$-subsets on the forward pass; I-MLE, which denotes the IMLE gradient estimator (Niepert et al., 2021), where *approximate* samples are obtained using perturb-and-map (PAM) on the forward pass, approximating the marginals using PAM samples on the backward pass; and score function estimator, denoted *SFE*.

We tease apart SIMPLE's improvements by comparing three different flavors: SIMPLE-F, which only uses *exact* sampling, falling back to estimating the marginals using *exact* samples; SIMPLE-B, which uses *exact* marginals on the backward pass with *approximate* PAM samples on the forward pass; and SIMPLE, coupling *exact* samples on the forward pass with *exact* marginals on the backward pass.

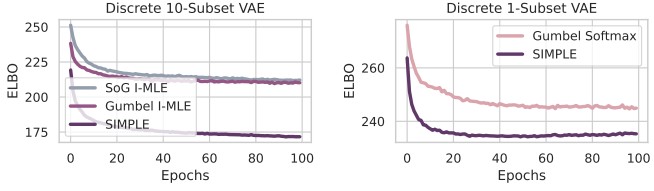

Figure 4: ELBO against # of epochs. (Left) Comparison of SIM-PLE against different flavors of IMLE on the 10-subset DVAE, and (Right) against ST Gumbel Softmax on the 1-subset DVAE.

**Algorithm 4** Entropy($\boldsymbol{\theta}, n, k$)

**Input:** The logits $\boldsymbol{\theta}$ of the distribution, the number of variables $n$, and the subset size $k$
**Output:** $H(\boldsymbol{z}) = -\mathbb{E}_{\boldsymbol{z} \sim p_{\boldsymbol{\theta}}(\boldsymbol{z}|\sum_i z_i = k)}[\log p(\boldsymbol{z})]$
$\quad h = \text{zeros}(n, k)$
$\quad$ **for** $i = k$ **to** $n$ **do**
$\quad\quad$ **for** $j = 0$ **to** $k$ **do**
$\quad\quad\quad$ // $p(z_i \mid \sum_{m=1}^{i} z_m = j)$
$\quad\quad\quad p = a[i-1, j-1] * p_{\theta_i}(z_i = 1)/a[i,j]$
$\quad\quad\quad$ // cf. proof of Prop. 3 in Appendix
$\quad\quad\quad h[i,j] = H_b(p) + p * h[i-1, j] +$
$\quad\quad\quad\quad (1-p) * h[i-1, j+1]$
$\quad$ **return** $h$

Our results are shown in Figure 1 As expected, we observe that *SFE* exhibits no bias, but high variance whereas *SoftSub* suffers from both bias and variance, due to the Gumbel noise injection into the samples to make them differentiable. We observe that I-MLE exhibits very high bias, as well as very low variance. This can be attributed to the PAM sampling, which in the case of $k$-subset distribution does not sample faithfully from the distribution, but is instead biased to sampling only the mode of the distribution. This also means that, by approximating the marginals using PAM samples, there is a lot less variance to our gradients. On to our SIMPLE gradient estimator, we see that it exhibits *less bias as well as less variance* compared to all the other gradient estimators. We also see that each estimated gradient is, on average, much more aligned with the exact gradient. To understand why that is, we compare SIMPLE, SIMPLE-F, and SIMPLE-B. As hypothesized, we observe that *exact sampling*, SIMPLE-F, reduces the bias, but increases the variance compared to I-MLE, this is since, unlike the PAM samples, our exact sample span the entire sample space. We also observe that, even compared to I-MLE, SIMPLE-B, reduces the variance by marginalizing over all possible samples.

## 5.2 DISCRETE VARIATIONAL AUTO-ENCODER

Next, we test our SIMPLE gradient estimator in the $k$-subset discrete variational auto-encoder (DVAE) setting, where the latent variables model a probability distribution over $k$-subsets, and has a dimensionality of 20. Similar to prior work (Jang et al., 2017; Niepert et al., 2021), the encoding and decoding functions of the VAE consist of three dense layers (encoding: 512-256-20x20; decoding: 256-512-784). The DVAE is trained to minimize the sum of reconstruction loss and KL-divergence of the $k$-subset distribution and the constrained uniform distribution, known as the ELBO, on MNIST.

In prior work, the KL-divergence was approximated using the unconditional marginals, obtained simply through a Softmax layer. Instead we show that the KL-divergence between the $k$-subset distribution and the uniform distribution can be computed exactly. First note that, through simple algebraic manipulations, the KL-divergence between the $k$-subset distribution and the constrained uniform distribution can be rewritten as the sum of negative entropy, $-H(\boldsymbol{z})$, where $\boldsymbol{z} \sim p_{\boldsymbol{\theta}}(\boldsymbol{z} \mid \sum_i z_i = k)$ and log the number of $k$-subsets, $\log \binom{n}{k}$ (see appendix for details), reducing the hardness of computing the KL-divergence, to computing the entropy of a $k$-subset distribution, for which Algorithm 4 gives a tractable algorithm. Intuitively, the uncertainty in the distribution over a sequence of length $n$, $k$ of which are true, decomposes as the uncertainty over $Z_n$, and the average of the uncertainties over the remainder of the sequence. We refer the reader to the appendix for the proof of the below proposition.

**Proposition 3.** *Let* Entropy *be defined as in Algorithm 4. Given variables,* $Z_1, \cdots, Z_n$, *and a $k$-subset distribution* $p_{\boldsymbol{\theta}}(\boldsymbol{z} \mid \sum_i z_i = k)$, *Algorithm 4 computes entropy of* $p_{\boldsymbol{\theta}}(\boldsymbol{z} \mid \sum_i z_i = k)$.

We plot the loss ELBO against the number of epochs, as seen in Figure 4. We compared against I-MLE using sum-of-gamma noise as well as Gumbel noise for PAM sampling, on the 10-subset DVAE, and against ST Gumbel Softmax on the 1-subset DVAE. We observe a *significantly* lower loss on the test set on the 10-subset DVAE, partly attributable to the exact ELBO computation, but also on the 1-subset DVAE compared to ST Gumbel Softmax, where the sole difference is the backward pass.

## 5.3 LEARNING TO EXPLAIN

The BEERADVOCATE dataset (McAuley et al., 2012) consists of free-text reviews and ratings for 4 different aspects of beer: appearance, aroma, palate, and taste. The training set has 80k reviews for the aspect APPEARANCE and 70k reviews for all other aspects. In addition to the ratings for all reviews, each sentence in the test set contains annotations of the words that best describe the review

| Method | Appearance | | Palate | | Taste | |
|---|---|---|---|---|---|---|
| | Test MSE | Precision | Test MSE | Precision | Test MSE | Precision |
| SIMPLE (Ours) | **2.35 ± 0.28** | **66.81 ± 7.56** | **2.68 ± 0.06** | **44.78 ± 2.75** | **2.11 ± 0.02** | **42.31 ± 0.61** |
| L2X (t = 0.1) | 10.70 ± 4.82 | 30.02 ± 15.82 | 6.70 ± 0.63 | **50.39 ± 13.58** | 6.92 ± 1.61 | 32.23 ± 4.92 |
| SoftSub (t = 0.5) | **2.48 ± 0.10** | 52.86 ± 7.08 | 2.94 ± 0.08 | 39.17 ± 3.17 | 2.18 ± 0.10 | **41.98 ± 1.42** |
| I-MLE (τ = 30) | **2.51 ± 0.05** | **65.47 ± 4.95** | 2.96 ± 0.04 | 40.73 ± 3.15 | 2.38 ± 0.04 | **41.38 ± 1.55** |

Table 2: Results for three aspects with $k = 10$: test MSE and subset precision, both $\times 100$

| Method | $k = 5$ | | $k = 10$ | | $k = 15$ | |
|---|---|---|---|---|---|---|
| | Test MSE | Precision | Test MSE | Precision | Test MSE | Precision |
| SIMPLE (Ours) | **2.27 ± 0.05** | **57.30 ± 3.04** | **2.23 ± 0.03** | **47.17 ± 2.11** | 3.20 ± 0.04 | **53.18 ± 1.09** |
| L2X (t = 0.1) | 5.75 ± 0.30 | 33.63 ± 6.91 | 6.68 ± 1.08 | 26.65 ± 9.39 | 7.71 ± 0.64 | 23.49 ± 10.93 |
| SoftSub (t = 0.5) | 2.57 ± 0.12 | **54.06 ± 6.29** | 2.67 ± 0.14 | 44.44 ± 2.27 | **2.52 ± 0.07** | 37.78 ± 1.71 |
| I-MLE (τ = 30) | 2.62 ± 0.05 | **54.76 ± 2.50** | 2.71 ± 0.10 | **47.98 ± 2.26** | 2.91 ± 0.18 | 39.56 ± 2.07 |

Table 3: Results for aspect Aroma: test MSE and subset precision, both $\times 100$, for $k \in \{5, 10, 15\}$.

score with respect to the various aspects. We address the problem introduced by the L2X paper (Chen et al., 2018) of learning a $k$-subset distribution over words that best explain a given rating. We follow the architecture suggested in the L2X paper, consisting of four convolutional and one dense layer.

We compare to relaxation-based baselines L2X (Chen et al., 2018) and SoftSub (Xie & Ermon, 2019) as well as to I-MLE which uses perturb-and-MAP to both compute an approximate sample in the forward pass and to estimate the marginals. Prior work has shown that the straight-through estimator (STE) did not work well and we omit it here. We used the standard hyperparameter settings of Chen et al. (2018) and choose the temperature parameter $t \in \{0.1, 0.5, 1.0, 2.0\}$ for all methods. We used the standard Adam settings and trained separate models for each aspect using MSE as point-wise loss $\ell$. Table 3 lists results for $k \in \{5, 10, 15\}$ for the AROMA aspect. The mean-squared error (MSE) of SIMPLE is almost always lower and its subset precision never significantly exceeded by those of the baselines. Table 2 shows results on the remaining aspects Appearance, Palate, and Taste for $k = 10$.

## 5.4 SPARSE LINEAR REGRESSION

Given a library of feature functions, the task of sparse linear regression aims to learn from data which feature subset best describes the nonlinear partial differential equation (PDE) that the data are sampled from. We propose to tackle this task by learning a $k$-subset distribution over the feature functions. During learning, we first sample from the $k$-subset distribution to decide which feature function subset to choose. With $k$ chosen features, we perform linear regression to learn the coefficients of the features from data, and then update the $k$-subset distribution logit parameters by minimizing RMSE.

To test our proposed approach, we follow the experimental setting in PySINDy (de Silva et al., 2020; Kaptanoglu et al., 2022) and use the dataset collected by PySINDy where the samples are collected from the Kuramoto–Sivashinsky (KS) equation, a fourth-order nonlinear PDE known for its chaotic behavior. This PDE takes the form $v_t = -v_{xx} - v_{xxxx} - vv_x$, which can be seen as a linear combination of feature functions $\mathcal{V} = \{v_{xx}, v_{xxxx}, vv_x\}$ with the coefficients all set to a value of $-1$. At test time, we use the MAP estimation of the learned $k$-subset distribution to choose the $k$ feature functions. For $k = 3$, our proposed method achieves the same performance as the state-of-the-art solver on this task, PySINDy. It identifies the KS PDE from data by choosing exactly the ground truth feature function subset $\mathcal{V}$, obtaining an RMSE of $0.00622$ after applying linear regression on $\mathcal{V}$.

## 6 COMPLEXITY ANALYSIS

In Proposition 1, we prove that computing the marginal probability of the exactly-$k$ constraint can be done tractably in time $\mathcal{O}(nk)$. In the context of deep learning, we often care about vectorized com-

plexity. We demonstrate an optimized algorithm achieving a vectorized complexity $\mathcal{O}(\log k \log n)$, assuming perfect parallelization. The optimization is possible by computing the marginal probability in a divide-and-conquer way: it partitions the variables into two subsets and compute their marginals respectively such that the complexity $\mathcal{O}(n)$ is reduced to $\mathcal{O}(\log n)$; the summation over the $k$ terms also has its complexity reduced to $\mathcal{O}(\log k)$ in a similar manner. We refer the readers to Algorithm 5 in Appendix for the optimized algorithm. We further modify Algorithm 2 to perform divide-and-conquer such that sampling $k$-subsets achieves a vectorized complexity being $\mathcal{O}(\log n)$, shown as Algorithm 6 in the Appendix. As a comparison, SoftSub (Xie & Ermon, 2019) has its complexity to be $\mathcal{O}(nk)$ due to the relaxed top-k operation and its vectorized complexity to be $\mathcal{O}(k \log n)$ stemming from the fact that softmax layers need $\mathcal{O}(\log n)$ rounds of communication for normalization.

## 7 RELATED WORK

There is a large body of work on gradient estimation for categorical random variables. Maddison et al. (2017); Jang et al. (2017) propose the Gumbel-softmax distribution (named the concrete distribution by the former) to relax categorical random variables. For more complex distributions, such as the $k$-subset distribution which we are concerned with in this paper, existing approaches either use the straight-through and score function estimators or propose tailor-made relaxations (see for instance Kim et al. (2016); Chen et al. (2018); Grover et al. (2018)). We directly compare to the score function and straight-through estimator as well as the tailored relaxations of Chen et al. (2018); Grover et al. (2018) and show that we are competitive and obtain a lower bias and/or variance than these other estimators. Tucker et al. (2017); Grathwohl et al. (2018) develop parameterized control variates based on continuous relaxations for the score-function estimator. Lastly, Paulus et al. (2020) offers a comprehensible work on relaxed gradient estimators, deriving several extensions of the softmax trick. All of the above works, ours included, assume the independence of the selected items, beyond there being $k$ of them. That is with the exception of Paulus et al. (2020) which make use of a relaxation using pairwise embeddings, but do not make their code available. We leave that to future work.

A related line of work has developed and analyzed sparse variants of the softmax function, motivated by their potential computational and statistical advantages. Representative examples are Blondel et al. (2020a); Peters et al. (2019); Correia et al. (2019); Martins & Astudillo (2016). SparseMAP (Niculae et al., 2018) has been proposed in the context of structured prediction and latent variable models, also replacing the softmax with a sparser distribution. LP-SparseMAP (Niculae & Martins, 2020) is an extension that uses a relaxation of the optimization problem rather than a MAP solver. Sparsity can also be exploited for efficient marginal inference in latent variable models (Correia et al., 2020). Contrary to our work, they cannot control the sparsity level exactly through a $k$-subset constraint or guarantee a sparse output. Also, we aim at cases where samples in the forward pass are required.

Integrating specialized discrete algorithms into neural networks is a line of research with increasing popularity. Examples are sorting algorithms (Cuturi et al., 2019; Blondel et al., 2020b; Grover et al., 2018), ranking (Rolinek et al., 2020; Kool et al., 2019), dynamic programming (Mensch & Blondel, 2018; Corro & Titov, 2019), and solvers for combinatorial optimization problems Berthet et al. (2020); Rolínek et al. (2020); Shirobokov et al. (2020); Niepert et al. (2021); Minervini et al. (2023); Zeng et al. or even probabilistic circuits over structured output spaces (Ahmed et al., 2022; Blondel, 2019). There has also been work on making common programming language expression such as conditional statements, loops, and indexing differentiable through relaxations (Petersen et al., 2021). Xie et al. (2020) proposes optimal transport as a way to obtain differentiable sorting methods for top-$k$ classification. In contrast, we focus on the $k$-subset sampling problem and provide exact discrete sampling and marginal inference algorithms, obtaining a gradient estimator for the $k$-subset distribution with a favorable bias-variance trade-off.

## 8 CONCLUSION

We introduced a gradient estimator for the $k$-subset distribution which replaces *relaxed* and *approximate* sampling on the forward pass with *exact* sampling. It sidesteps the non-differentiable nature of discrete sampling by estimating the gradients as a function of our distribution's marginals, for which we prove a simple characterization, showing that we can compute them exactly and efficiently. We demonstrated improved empirical results on a number of tasks: L2X, DVAEs, and sparse regression.

## ACKNOWLEDGMENTS

We would like to thank all the reviewers whose insightful comments helped improve this work. We would especially like to thank reviewers for pointing out that we can forgo the method of finite differences in favor of computing the directional derivative, alleviating the need for $\lambda$, and enabling us to compute exact derivative along the downstream gradient. This work was funded in part by the DARPA Perceptually-enabled Task Guidance (PTG) Program under contract number HR00112220005, NSF grants #IIS-1943641, #IIS-1956441, #CCF-1837129, Samsung, CISCO, a Sloan Fellowship, and by Deutsche Forschungsgemeinschaft under Germany's Excellence Strategy - EXC 2075. ZZ is supported by an Amazon Doctoral Student Fellowship.

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

## A    PROOFS

**Proposition 2.**    *Let $p_{\boldsymbol{\theta}}(\sum_i z_i = k)$ be the probability of exactlty-k of the unconstrained distribution parameterized by log probabilities $\boldsymbol{\theta}$. For every $Z_i$, the conditional marginal is*

$$p_{\boldsymbol{\theta}}\left(z_i \mid \sum_j z_j = k\right) = \frac{\partial}{\partial \theta_i} \log p_{\boldsymbol{\theta}}(\sum_j z_j = k). \tag{10}$$

*Proof.*  We first rewrite the marginal $p_{\boldsymbol{\theta}}(\sum_i z_i = k)$ into a summation as the probability for all possible events by definition as follows.

$$p_{\boldsymbol{\theta}}(\sum_j z_j = k) = \sum_{\boldsymbol{z}:\sum_j z_j = k} \prod_{j:z_j=1} \exp(\theta_j) \prod_{j:z_j=0}(1 - \exp(\bar{\theta}_j)) \tag{11}$$

Here we assume that the probability of $z_j = 0$ is a constant term w.r.t. parameter $\theta_j$, i.e., $\frac{\partial}{\partial \theta_j}(1 - \exp(\bar{\theta}_j)) = 0$.[*] Further, the derivative of $p_{\boldsymbol{\theta}}(\sum_j z_j = k)$ w.r.t. $\theta_i$ is as follows,

$$\begin{aligned}
\frac{\partial}{\partial \theta_i} p_{\boldsymbol{\theta}}(\sum_j z_j = k) &= \frac{\partial}{\partial \theta_i} \sum_{\boldsymbol{z}:\sum_j z_j = k \wedge z_i = 1} \prod_{j:z_j=1} \exp(\theta_j) \prod_{j:z_j=0}(1 - \exp(\bar{\theta}_j)) \\
&= \frac{\partial}{\partial \theta_i} \exp(\theta_i) \sum_{\boldsymbol{z}:\sum_j z_j = k \wedge z_i = 1} \prod_{j:z_j=1, j \neq i} \exp(\theta_j) \prod_{j:z_j=0}(1 - \exp(\bar{\theta}_j)) \\
&= \exp(\theta_i) \sum_{\boldsymbol{z}:\sum_j z_j = k \wedge z_i = 1} \prod_{j:z_j=1, j \neq i} \exp(\theta_j) \prod_{j:z_j=0}(1 - \exp(\bar{\theta}_j)) \\
&= p_{\boldsymbol{\theta}}(\sum_j z_j = k \wedge z_i = 1),
\end{aligned}$$

where the first equality holds since terms corresponding to $z_i \neq 1$ has their derivative to be zero w.r.t. $\theta_i$. It further holds that

$$\begin{aligned}
\frac{\partial}{\partial \theta_i} \log p_{\boldsymbol{\theta}}(\sum_j z_j = k) &= \frac{\frac{\partial}{\partial \theta_i} p_{\boldsymbol{\theta}}(\sum_j z_j = k)}{p_{\boldsymbol{\theta}}(\sum_j z_j = k)} \\
&= \frac{p_{\boldsymbol{\theta}}(\sum_j z_j = k \wedge z_i = 1)}{p_{\boldsymbol{\theta}}(\sum_j z_j = k)} \\
&= p_{\boldsymbol{\theta}}\left(z_i \mid \sum_j z_j = k\right)
\end{aligned}$$

which finishes our proof.    □

**Proposition 3.**    *Let `Entropy` be defined as in Algorithm 4. Given variables $Z_1, \cdots, Z_n$ and a k-subset distribution $p_{\boldsymbol{\theta}}(\boldsymbol{z} \mid \sum_i z_i = k)$ parameterized by $\boldsymbol{\theta}$, Algorithm 4 computes entropy of $p_{\boldsymbol{\theta}}(\mathbf{z} \mid \sum_i z_i = k)$.*

*Proof.*  In a slight abuse of notation, let $z_n$ denote $z_n = 1$, and let $\bar{z}_n$ denote $z_n = 0$. Furthermore, we denote by $\sigma_n^k, \sigma_{n-1}^k$ and $\sigma_{n-1}^{k-1}$ the events $\sum_{i=0}^n = k, \sum_{i=0}^{n-1} = k$ and $\sum_{i=0}^{n-1} = k-1$, respectively.

The entropy of the $k$-subset distribution is given by

$$H(\boldsymbol{Z}) = -\mathbb{E}_{\boldsymbol{z} \sim p_{\boldsymbol{\theta}}(\boldsymbol{z} \mid \sigma_n^k)} [\log p(\boldsymbol{z})] = -\sum_{\boldsymbol{z}:\sigma_n^k} p_{\boldsymbol{\theta}}(\boldsymbol{z} \mid \sigma_n^k) \log p_{\boldsymbol{\theta}}(\boldsymbol{z} \mid \sigma_n^k)$$

We start by simplifying the expression for $p_{\boldsymbol{\theta}}(\boldsymbol{z} \mid \sigma_n^k)$, where, by the chain rule , the above is

$$\sum_{\boldsymbol{z}:\sigma_n^k} p_{\boldsymbol{\theta}}(\bar{z}_n \mid \sigma_n^k) \cdot p_{\boldsymbol{\theta}}(\sigma_{n-1}^k \mid \sigma_n^k, \bar{z}_n) + p_{\boldsymbol{\theta}}(z_n \mid \sigma_n^k) \cdot p_{\boldsymbol{\theta}}(\sigma_{n-1}^{k-1} \mid \sigma_n^k, z_n)$$

---

[*]In practice, this can be easily implemented. For example, in framework `Tensorflow`, it can be done by setting `tf.stop_gradients`.

Plugging the above in the expression for the entropy, distributing the sum over the product, we get

$$
= -\sum_{\boldsymbol{z}:\sigma_n^k} p_{\boldsymbol{\theta}}(\bar{z}_n \mid \sigma_n^k) \cdot p_{\boldsymbol{\theta}}(\sigma_{n-1}^k \mid \sigma_n^k, \bar{z}_n)
$$
$$
\cdot \log \left[ p_{\boldsymbol{\theta}}(\bar{z}_n \mid \sigma_n^k) \cdot p_{\boldsymbol{\theta}}(\sigma_{n-1}^k \mid \sigma_n^k, \bar{z}_n) + p_{\boldsymbol{\theta}}(z_n \mid \sigma_n^k) \cdot p_{\boldsymbol{\theta}}(\sigma_{n-1}^{k-1} \mid \sigma_n^k, z_n) \right]
$$
$$
+ p_{\boldsymbol{\theta}}(z_n \mid \sigma_n^k) \cdot p_{\boldsymbol{\theta}}(\sigma_{n-1}^{k-1} \mid \sigma_n^k, z_n)
$$
$$
\cdot \log \left[ p_{\boldsymbol{\theta}}(\bar{z}_n \mid \sigma_n^k) \cdot p_{\boldsymbol{\theta}}(\sigma_{n-1}^k \mid \sigma_n^k, \bar{z}_n) + p_{\boldsymbol{\theta}}(z_n \mid \sigma_n^k) \cdot p_{\boldsymbol{\theta}}(\sigma_{n-1}^{k-1} \mid \sigma_n^k, z_n) \right],
$$

where, since the two events $\bar{z}_n$ and $z_n$ are mutually exclusive, we can simplify the above to

$$
-\sum_{\boldsymbol{z}:\sigma_n^k} p_{\boldsymbol{\theta}}(\bar{z}_n \mid \sigma_n^k) \cdot p_{\boldsymbol{\theta}}(\sigma_{n-1}^k \mid \sigma_n^k, \bar{z}_n) \cdot \log \left[ p_{\boldsymbol{\theta}}(\bar{z}_n \mid \sigma_n^k) \cdot p_{\boldsymbol{\theta}}(\sigma_{n-1}^k \mid \sigma_n^k, \bar{z}_n) \right]
$$
$$
+ p_{\boldsymbol{\theta}}(z_n \mid \sigma_n^k) \cdot p_{\boldsymbol{\theta}}(\sigma_{n-1}^{k-1} \mid \sigma_n^k, z_n) \cdot \log \left[ p_{\boldsymbol{\theta}}(z_n \mid \sigma_n^k) \cdot p_{\boldsymbol{\theta}}(\sigma_{n-1}^{k-1} \mid \sigma_n^k, z_n) \right].
$$

Expanding the logarithms, rearranging terms, and using that conditional probabilities sum to 1 we get

$$
p_{\boldsymbol{\theta}}(\bar{z}_n \mid \sigma_n^k) \log p_{\boldsymbol{\theta}}(\bar{z}_n \mid \sigma_n^k) + p_{\boldsymbol{\theta}}(z_n \mid \sigma_n^k) \log p_{\boldsymbol{\theta}}(z_n \mid \sigma_n^k)
$$
$$
+ p_{\boldsymbol{\theta}}(\bar{z}_n \mid \sigma_n^k) \cdot p_{\boldsymbol{\theta}}(\sigma_{n-1}^k \mid \sigma_n^k, \bar{z}_n) \log p_{\boldsymbol{\theta}}(\sigma_{n-1}^k \mid \sigma_n^k, \bar{z}_n)
$$
$$
+ p_{\boldsymbol{\theta}}(z_n \mid \sigma_n^k) \cdot p_{\boldsymbol{\theta}}(\sigma_{n-1}^{k-1} \mid \sigma_n^k, z_n) \log p_{\boldsymbol{\theta}}(\sigma_{n-1}^{k-1} \mid \sigma_n^k, z_n)
$$
$$
= -\mathbb{E}_{z_n \sim p_{\boldsymbol{\theta}}(z_n \mid \sigma_n^k)} \left[ -\log p_{\boldsymbol{\theta}}(z_n \mid \sigma_n^k) \right] + \mathbb{E}_{z_n \sim p_{\boldsymbol{\theta}}(z_n \mid \sigma_n^k)} \left[ H(\boldsymbol{Z}_{:n-1} \mid \sigma_n^k, z_n) \right]
$$
$$
= H_b(Z_n \mid \sigma_n^k) + \mathbb{E}_{z_n \sim p_{\boldsymbol{\theta}}(z_n \mid \sigma_n^k)} \left[ H(\boldsymbol{Z}_{:n-1} \mid \sigma_n^k, z_n) \right].
$$

That is, simply stated, the entropy of the $k$-subset distribution decomposes as the entropy of the *constrained* distribution over $Z_n$, and average entropy of the distribution on the remaining variables.

As the base case, the entropy of the $k$-subset distribution when $k = n$ is 0; there is only one way in which to pick to choose $n$ of $n$ variables, and the $k$-subset distribution is therefore deterministic. $\square$

## B  OPTIMIZED ALGORITHMS

Algorithm 5 is the optimized version of Algorithm 1, both of which compute the marginal probability of the exactly-$k$ constraint. Algorithm 6 is the optimized version of Algorithm 2, both of which sample faithfully from the $k$-subset distribution.

---

**Algorithm 5** $\texttt{PrExactlyk}(\boldsymbol{\theta}, l, u, k)$

**Input**: The logits $\boldsymbol{\theta}$ of the distribution, range of variable indices $[l, u]$, and the subset size $k$
**Output**: The exact marginal probability of variables summing up to $k$, $P(\sum_{i=l}^u X_i = k)$
  **if** $l > u$ **then return** 0
  **if** $l = u$ **then return** $p_{\boldsymbol{\theta}}(X_l = k)$
  **for** $m = 0$ to $k$ **do**
    $p_m = \texttt{PrExactlyk}(\boldsymbol{\theta}, l, \lfloor u/2 \rfloor, m) *$
      $\texttt{PrExactlyk}(\boldsymbol{\theta}, \lfloor u/2 \rfloor + 1, u, k - m)$
  **return** $\sum_{m=0}^k p_m$

---

**Algorithm 6** $\texttt{Sample}(\boldsymbol{\theta}, l, u, k)$

**Input**: The logits $\boldsymbol{\theta}$ of the distribution, range of variable indices $[l, u]$, and the subset size $k$
**Output**: A sample $\boldsymbol{z} = (z_1, \ldots, z_n)$ from $p_{\boldsymbol{\theta}}(\boldsymbol{z} \mid \sum_i z_i = k)$
  define $p(x = m) = p_m, m = 0, \cdots, k$
  // with $p_m$ as defined in Algorithm 5
  sample $m^*$ from $p$
  $\boldsymbol{z}_{l:\lfloor u/2 \rfloor} = \texttt{Sample}(\boldsymbol{\theta}, l, \lfloor u/2 \rfloor, m^*)$
  $\boldsymbol{z}_{\lfloor u/2 \rfloor+1:u} = \texttt{Sample}(\boldsymbol{\theta}, \lfloor u/2 \rfloor + 1, u, k - m^*)$
  **return** $\texttt{Concat}(\boldsymbol{z}_{l:\lfloor u/2 \rfloor}, \boldsymbol{z}_{\lfloor u/2 \rfloor+1:u})$

---

## C  EXPERIMENTAL DETAILS

### C.1  SYNTHETIC EXPERIMENTS

In this experiment we analyzed the behavior of various discrete gradient estimators for the $k$-subset distribution. We were interested in three different metrics: the bias of the the gradients estimators, the variance of the gradient estimators, as well as the average deviation of each estimated gradient from the exact gradient. We used cosine distance, defined as $1-$ cosine similarity as our measure of

distance, as we typically care about the direction, not the magnitude of the gradient; the latter can be recovered using an appropriate learning rate. Following Niepert et al. (2021), we chose a tractable 5-subset distribution, where $n = 10$, and were therefore limited to $\binom{10}{5} = 252$ possible subsets. We set the loss to $L(\boldsymbol{\theta}) = \mathbb{E}_{z \sim p_{\boldsymbol{\theta}}(\mathbf{z} | \sum_i z_i = k)}[\|\mathbf{z} - \mathbf{b}\|^2]$, where $\mathbf{b}$ is the groundtruth logits sampled from $\mathcal{N}(0, \mathbf{I})$. We used a sample size of 10000 to estimate each of our metrics.

## C.2 Discrete Variational Auto-Encoder

We tested our SIMPLE gradient estimator in the discrete $k$-subset Variational Auto-Encoder (VAE) setting, where the latent variables model a probability distribution over $k$-subsets, and has a dimensionality of 20. The experimental setup is similar to those used in prior work on the Gumbel softmax tricks (Jang et al., 2017) and IMLE (Niepert et al., 2021) The encoding and decoding functions of the VAE consist of three dense layers (encoding: 512-256-20x20; decoding: 256-512-784). As is commonplace in discrete VAEs, the loss is the sum of the reconstruction loss (binary cross-entropy loss on output pixels) and KL divergence of the $k-$subset distribution and the uniform distribution, known as the evidence lower bound, or the ELBO. The task being to learn a *sparse* generative model of MNIST. As in prior work, we use a batch size of 100 and train for 100 epochs, plotting the test loss after each epoch. We use the standard Adam settings in Tensorflow 2.x, and do not employ any learning rate scheduling. The encoder network consists of an input layer with dimension 784 (we flatten the images), a dense layer with dimension 512 and ReLu activation, a dense layer with dimension 256 and ReLu activation, and a dense layer with dimension $400(20 \times 20)$ which outputs $\boldsymbol{\theta}$ and no non-linearity SIMPLE takes $\boldsymbol{\theta}$ as input and outputs a discrete latent code of size $20 \times 20$. The decoder network, which takes this discrete latent code as input, consists of a dense layer with dimension 256 and ReLu activation, a dense layer with dimension 512 and ReLu activation, and finally a dense layer with dimension 784 returning the logits for the output pixels. Sigmoids are applied to these logits and the binary cross-entropy loss is computed. To obtain the best performing model of each of the compared methods, we performed a grid search over the learning rate in the range $[1 \times 10^{-3}, 5 \times 10^{-4}]$, $\lambda$ in the range $[1 \times 10^{-3}, 1 \times 10^{-2}, 1 \times 10^{-1}, 1 \times 10^0, 1 \times 10^1, 1 \times 10^2, 1 \times 10^3]$, and for SoG I-MLE, the temparature $\tau$ in the range $[1 \times 10^{-1}, 1 \times 10^0, 1 \times 10^1, 1 \times 10^2]$

We will now present a formal proof on how to compute the KL-divergence between the $k$-subset distribution and a uniform distribution tractably and exactly.

**Proposition 4.** *Let $p_{\boldsymbol{\theta}}(\mathbf{z} | \sum_i z_i = k)$ be a $k$-subset distribution parameterized by $\boldsymbol{\theta}$ and $\mathcal{U}(\boldsymbol{z})$ be a uniform distribution on the constrained space $\mathcal{C} = \{\boldsymbol{z} | \sum_i z_i = k\}$. Then the KL-divergence between distribution $p_{\boldsymbol{\theta}}(\mathbf{z} | \sum_i z_i = k)$ and $\mathcal{U}(\boldsymbol{z})$ can be computed by*

$$D(p_{\boldsymbol{\theta}}(\mathbf{z} | \sum_i z_i = k) \| \mathcal{U}(\boldsymbol{z})) = -H(\boldsymbol{z}) + \log \binom{n}{k},$$

*where $H$ denote the entropy of distribution $p_{\boldsymbol{\theta}}(\mathbf{z} | \sum_i z_i = k)$.*

*Proof.* By the definition of KL divergence, it holds that

$$D(p_{\boldsymbol{\theta}}(\mathbf{z} | \sum_i z_i = k) \| \mathcal{U}(\boldsymbol{z}))$$

$$= \sum_{\boldsymbol{z} \in \mathcal{C}} p_{\boldsymbol{\theta}}(\mathbf{z} | \sum_i z_i = k) \cdot \log \frac{p_{\boldsymbol{\theta}}(\mathbf{z} | \sum_i z_i = k)}{U(\boldsymbol{z})}$$

$$= (\sum_{\boldsymbol{z} \in \mathcal{C}} p_{\boldsymbol{\theta}}(\mathbf{z} | \sum_i z_i = k) \log p_{\boldsymbol{\theta}}(\mathbf{z} | \sum_i z_i = k)) - \sum_{\boldsymbol{z} \in \mathcal{C}} p_{\boldsymbol{\theta}}(\mathbf{z} | \sum_i z_i = k) \log U(\boldsymbol{z})$$

$$= -H(\boldsymbol{z}) + \log \binom{n}{k}.$$

The last equality holds since $U(\boldsymbol{z}) \equiv 1/\binom{n}{k}$. $\qquad\square$

## C.3 Learning to Explain

The BEERADVOCATE dataset (McAuley et al., 2012) consists of free-text reviews and ratings for 4 different aspects of beer: appearance, aroma, palate, and taste. The training set has 80k reviews for

the aspect APPEARANCE and 70k reviews for all other aspects. The maximum review length is 350 tokens. We follow Niepert et al. (2021) in computing 10 different evenly sized validation/test splits of the 10k held out set and compute mean and standard deviation over 10 models, each trained on one split. In addition to the ratings for all reviews, each sentence in the test set contains annotations of the words that best describe the review score with respect to the various aspects. Following the experimental setup of recent work (Paulus et al., 2020; Niepert et al., 2021), we address the problem introduced by the L2X paper (Chen et al., 2018) of learning a $k$-subset distribution over words that best explain a given aspect rating. Subset precision was computed using a set of 993 annotated reviews. We use pre-trained word embeddings from Lei et al. (2016)[†] We use the standard neural network architecture from prior work Chen et al. (2018); Paulus et al. (2020) with 4 convolutional and one dense layer. This neural network outputs the parameters $\boldsymbol{\theta}$ of the $k$-subset distribution over $k$-hot binary latent masks with $k \in \{5, 10, 15\}$. We train for 20 epochs using the standard Adam settings in Tensorflow 2.x, and no learning rate schedule. We always evaluate the model with the best validation MSE among the 20 epochs.

---

[†]http://people.csail.mit.edu/taolei/beer/.

