# OpenReview forum: "SIMPLE: A Gradient Estimator for k-Subset Sampling"
_ICLR.cc/2023/Conference — ICLR 2023 poster_

### Official Review · Reviewer_AhEr · 2022-10-22

**Confidence:** 2
**Correctness:** 4
**Technical Novelty And Significance:** 2
**Empirical Novelty And Significance:** 2
**Recommendation:** 6

**Clarity, Quality, Novelty And Reproducibility:**

The paper provides a gradient estimator when the computation graph includes a k-subset sampling module.

**Strength And Weaknesses:**

* Strength

The paper provides a gradient estimator when the computation graph includes a k-subset sampling module.

* Weakness

Comparisons with other methods can be made more comprehensive.

**Summary Of The Paper:**

The paper provides a gradient estimator when the computation graph includes a k-subset sampling module.

**Summary Of The Review:**

The paper provides a gradient estimator when the computation graph includes a k-subset sampling module, which is new to me.

---

> ### Author Response · Authors · 2022-11-10
> **Response to Reviewer AhEr**
>
> We would like to thank the reviewer for their review. We will address your concerns below.
>
> [Comparison to other methods]
>
> We would like to reiterate that, in this work, our focus is a gradient estimator for sampling distribution over $k$ of $n$ elements. In our experiments, we compare against the most prominent estimators concerned with the same problem, namely: IMLE and its variants [3], SoftSub [2], as well as L2X [1], where applicable. Our comparisons are both qualitative (c.f. Section 5.1) as well as quantitative (c.f. Section 5.2 and 5.3), where we compare against all the above baselines in Section 5.3, on the canonical task of explaining reviews with the most likely $k$-subset of words, which was first introduced in [1].
>
> Moreover, when $k=1$, SIMPLE reduces to a gradient estimator for sampling a Categorical distribution, much like the problem setting Gumbel-Softmax is concerned with. We show that, qualitatively, for $k=1$, SIMPLE leads to an estimator with lower bias and variance compared to Gumbel-Softmax (c.f. Section 4 and Figure 3), and quantitatively on the task of generative learning of MNIST digits using Discrete VAEs, as in the original Gumbel Softmax paper (c.f. Section 5.2 and figure 4 (right) in our paper, and Section 4.2 in [4]). Such an improvement requires only a minimal amount of change to the implementation of the Gumbel-Softmax: we backpropagate through the original marginals, as opposed to the perturbed marginals.
>
>
> [The contributions are only marginally significant or novel.]
>
> We would like to reiterate our contributions: *We propose an algorithm for exact $k$-subset sampling*, which we use on the forward pass to sample discretely from the latent space. On the backward pass, we make use of the straight-through estimator, parameterizing the gradient w.r.t the samples as the gradient w.r.t the marginals. *We derive the gradient of loss w.r.t the samples as the derivative of the marginals of the $k$-subset distribution in the direction of the downstream gradient*. Computing the marginals of the $k$-subset distribution might initially seem daunting. *We derive the marginals as the partial derivative of the probability of the exactly-k constraint*. This reduces the problem to computing the probability of exactly-k. *We derive a tractable algorithm for computing the probability of exactly-k, using backprop to compute the marginals of the $k$-subset distribution*.
>
> Regarding the significance of the proposed estimator, we have showcased several applications of $k$-subset sampling: learning discrete generative models e.g. DVAEs, learning to explain by selecting the most likely explanation from a distribution over all possible explanations, as well as sparse regression, where we’re interested in ignoring nuisance features when training our model. Not to mention the wide applicability of the Gumbel Softmax throughout the deep learning literature.
>
> Hopefully that answers the reviewer’s concerns regarding empirical comparisons, as well as the significance and novelty of the proposed approach, as we hope to move towards full acceptance.
>
> References:
>
> [1] Jianbo Chen, Le Song, Martin Wainwright, and Michael Jordan. Learning to explain: An information- theoretic perspective on model interpretation. In Proceed- ings of the 35th International Conference on Machine Learning.
>
> [2] Sang Michael Xie and Stefano Ermon. Reparameterizable subset sampling via continuous relaxations. In Proceedings of the Twenty-Eighth International Joint Conference on Artificial Intelligence.
>
> [3] Mathias Niepert, Pasquale Minervini, and Luca Franceschi. Implicit mle: Backpropagating through discrete exponential family distributions. Advances in Neural Information Processing Systems, 34, 2021.
>
> [4] Eric Jang, Shixiang Gu, and Ben Poole. Categorical reparameterization with gumbel-softmax. 5th International Conference on Learning Representations.

---

### Official Review · Reviewer_6jdF · 2022-10-24

**Confidence:** 4
**Correctness:** 3
**Technical Novelty And Significance:** 2
**Empirical Novelty And Significance:** 1
**Recommendation:** 3

**Clarity, Quality, Novelty And Reproducibility:**

The clarity of the paper is the major concern.

- It seems that the details are missing often, and a lot of parts are written informal way.

- It seems that the details are missing, and a lot of parts are written in informal way.

- For example, writing .detach() in the middle of the text seems inappropriate.

- What is jvp?

- The authors use terminology and abbreviation without consistency.

- What is SoG I-MLE?

- Which hyper-parameter values are used in Figure 1?

- The same problem happens in Figure 3, also. These make those experiments looks unfair and hard to compare the proposed model against baselines.

- Also, I suggest placing the proofs of the propositions in the appendix, and enhancing the experiment settings, for example in Section 5.4.

- For the case when k=1, there are lots of baselines which can be compared in the synthetic example and DVAE experiment. I’m surprising that the authors only compared to the Gumbel-Softmax in Figure 4.

- The paper seems to need more proof-reading, and it seems that the paper can be presented in the better shape.

Also, the code is not provided, which makes harder to understand paper and use the provided method.

Here are some questions to the authors.

- Could you compare the running time against the baselines, especially when k varies?

- I don’t understand the experiment setting in DVAE. Why does the decoder has 256-dim in the first decoder weight, while the last encoder weight has 20^2?


**Details Of Ethics Concerns:**

No ethics concerns.


**Strength And Weaknesses:**

The paper studies the exact gradient estimator for the n-choose-k distribution. The idea of this paper seems quiet novel. Also, it seems that the proposed method achieved better result in bias/variance, ELBO of DVAE, etc.

However, the paper is very hard to follow, not because of the difficulty of the paper, but due to the presentation of the paper.


**Summary Of The Paper:**

The paper suggest a k-subset sampling gradient estimator, namely SIMPLE. The proposed SIMPLE estimator can replace the reparameterization by relaxation with exact gradient estimation, and utilizes the gradient with respect to the exact conditional marginals in the backpropagtion. The authors conducted various experiments to demonstrate the performance of SIMPLE estimator: synthetic example, discrete variational autoencoder, learning to explain, and sparse linear regression.


**Summary Of The Review:**

I agree with the methodology, but for some experiments, the explanation of the experimental method is insufficient, and the reliability of the results is insufficient. Also, the presentation of the paper difficult to understand. I therefore lean negatively on this paper.

---

> ### Author Response · Authors · 2022-11-10
> **Response to Reviewer 6jdF**
>
> Thanks for the feedback and we would like to address your concerns.
>
> [clarity]
>
> - The way we present our work include both intuitive explanations for accessibility and rigorous descriptions, including equations and theorems, proofs. We would like the reviewer to be more concise on which part of work that they think would need more details for them to understand.
> - The .detach() function is a key technique for implementing the differentiation through relaxed samples and is necessary for understanding how differently SIMPLE and ST Gumbel-Softmax work during the backpropagation.
> - jvp stands for Jacobian-vector product, or directional derivative, as explained right after Equation (7). We’ll put the abbreviation there for reference.
> - SoG I-MLE is a variation of the I-MLE gradient estimator proposed in Niepert et al. (2021) where the relaxed samples are obtained by applying a Sum-of-Gamma perturbation instead of the standard local Gumbel perturbations.
> - For the choice of hyperparameters, in Figure 1, the temperature of SoftSub is chosen from $[0.1, 0.5, 1.0, 2.0]$ and the lambdas of IMLE follows the ones in Niepert et al., (2021). In Figure 3, there is no tunable hyperparamters for Straight-Through Gumbel Softmax. We’ve uploaded the code for all experiments where all the hyperparameters are specified.
> - We would like to point out that SIMPLE is proposed as a gradient estimator for $k$-subset distribution and therefore our empirical evaluations mainly focus on the cases when $k > 1$ which are out of the reach of most gradient estimators such as Gumbel-Softmax. For $k = 1$ where the widely used Gumbel Softmax estimator is applicable, we find out that even though SIMPLE has similar scheme as Gumbel Softmax estimator, it is able to deliver estimations with lower bias and variance, indicating that our proposed SIMPLE can be a strong candidate for future applications.
>
> [code]
>
> We have uploaded the code for reproducing all the experiments to the supplementary materials.
>
> [empirical runtime]
>
> We provide the runtime of both SIMPLE and SoftSub for synthetic experiments: SIMPLE 3.40 ms ± 6.08 µs per loop and SoftSub 1.11 ms ± 13 µs per loop, as an average over seven runs. Their similar runtimes are consistent with our complexity analysis that both of them have non-vectorized complexity $O(nk)$. For vectorized complexity, we’ve shown that our algorithm can be optimized to achieve complexity $O(\log k \log n)$ assuming perfect parallelization; while the vectorized complexity of SoftSub can be optimized to at most $O(k \log n)$ where the factor $k$ is due to their use of the relaxed top-$k$ operation.
>
> [DVAE]
>
> The last layer of the encoder, with dimension being 20*20, is the discrete latent space over which the exactly-$k$ constraints are imposed. The output of this discrete latent space would then be the input to the first layer of the decoder with its dimension being 256.
>
>
> We hope this clears up any confusion and concerns, and if so, we’re hoping to turn the verdict into an accept.

---

> > ### Comment · Reviewer_6jdF · 2022-11-19
> > **Response to the authors**
> >
> > Thanks for your response.
> > I'm still not convinced with your experimental runtime of SIMPLE.
> > For example, one can plot $n$ versus $k$ plot for the runtime, and it seems that I'm not the only one who is interested in the runtime.
> > Also, me and other reviewer want to know the limitation of the proposed SIMPLE.

---

> > > ### Author Response · Authors · 2022-11-21
> > > **Response to Reviewer 6jdF**
> > >
> > > Thanks for engaging with us, albeit slightly past the deadline, so we are no longer able to revise our submission. Nonetheless, we are happy to see that most of your concerns have been addressed, apart from the runtime, which we hopefully manage to resolve below.
> > >
> > > Below is a comparison of the the run times of SoftSub (first entry in every cell) and SIMPLE (second entry in every cell, after the comma). All times are in milliseconds.
> > >
> > > |      |     k=5    |    k=10    |    k=15    |
> > > |:----:|:----------:|:----------:|:----------:|
> > > | n=10 | 5.27, 7.21 |      /     |      /     |
> > > | n=15 |  5.35, 9.9 | 7.85, 8.35 |      /     |
> > > | n=20 | 5.27, 10.4 | 9.14, 10.5 | 12.6, 10.6 |
> > >
> > > And here are a few more data points:
> > >
> > > For $n=40$ and  $k =25$:  20.1, 23.8
> > >
> > > For $n=80$, $k=50$:  32.7, 43.0
> > >
> > > For $n=100$, $k=80$: 39.5, 53.5
> > >
> > > Regarding the limitations of SIMPLE:
> > >
> > > In this paper, similar to all previous works on $k$-subset selection, we assume that the items are picked independently. We agree that relaxing the factorized distribution assumption, which assumes the absence of correlations between items included in the subset, beyond the fact that there’s $k$ of them, could be beneficial in some cases. For instance, one might expect that in the L2X task, the presence of a word such as “delicious” might correlate strongly with the presence of a word such as “good”. In the presence of such correlations, however, the $k$-subset distribution becomes intractable. We do think that this is an interesting problem, but a non-trivial one, nonetheless, that we look forward to addressing in our future work.
> > >
> > > We will add the following line to our related works section: “All of the above works, ours included, assume the independence of the selected items, beyond there being $k$ of them. That is with the exception of [1] which make use of a relaxation using pairwise embeddings, but do not make their code available. We view extending our work to distributions where the $k$ items are correlated as a very interesting future endeavor.
> > >
> > > We hope this lays to rest any concerns you might have regarding our paper, and changes your recommendation to accept.
> > >
> > > References:
> > >
> > > [1] Max B. Paulus, Dami Choi, Daniel Tarlow, Andreas Krause, and Chris J. Maddison. 2020. Gradient estimation with stochastic softmax tricks. In Proceedings of the 34th International Conference on Neural Information Processing Systems.

---

### Official Review · Reviewer_pM21 · 2022-10-24

**Confidence:** 3
**Correctness:** 2
**Technical Novelty And Significance:** 3
**Empirical Novelty And Significance:** 3
**Recommendation:** 6

**Clarity, Quality, Novelty And Reproducibility:**

The estimator seems novel, algorithms are well described, I have not found link to the code of experiments nor supplementary files with it.

**Strength And Weaknesses:**

Strength:
-- efficient and novel gradient estimator
-- experiments on synthetic and real data
-- rigorous theoretical analysis

Weaknesses:
-- choice of baselines in experiments seems odd, authors claim that their estimator is better then Gumbel softmax estimator but do not present comparison of it's quality on real data
-- improper comparison with Gumbel Softmax estimator, algorithm proposed by authors has asymptotic worse then of Gumbel SoftMax by a factor of k therefore I wonder whether their reduction in variance and bias will be significant over GS if we choose number of samples for it greater then 1 as authors did to match running time of their estimator.
-- it's not clear why GS and proposed estimator have bias at all -- GS based on log-likelihood trick should be unbiased and proposed estimator is claimed to exactly compute probabilities therefore it should be unbiased too (or am I missing something here?). Backprop cannot increase bias if it's absent since chain rule is basically reduces to application of linear mapping on unbiased estimate.


**Summary Of The Paper:**

The paper proposes novel gradient estimator for k-subset sampling. Authors justify their method with rigorous proofs and conduct experiments to support their claims. Experiments have major flaws that should be addressed.

**Summary Of The Review:**

The proposed sampler is novel while experimentation section of the paper is flawed and requires clarifications.

---

> ### Author Response · Authors · 2022-11-10
> **Response to Reviewer pM21**
>
> We would like to thank the reviewer for their feedback. In what follows, we hope to address any concerns you might have.
>
> Before attending to the specifics, we would like to emphasize that, while we draw a connection between our gradient estimator for the case $k=1$ and the Gumbel Softmax estimator, that is not what we are primarily interested in. Rather, we are interested in the case where $k >1$, and therefore, we have a distribution over subsets of $k$ elements.  For instance, the *set* of latent features that best correspond to an MNIST digit (c.f. Section 5.2), or the set of that best explain a given review (c.f. Section 5.3). Clearly then, the problem in question here is much more general than the one considered by Gumbel-Softmax, which only concerns itself with the estimating the gradient when sampling a categorical distribution, an uneasy feat due to the non-differentiable nature of discrete sampling.
>
> We will now address the individual concerns:
>
> [choice of baselines]
>
> We compare against the most prominent gradient estimators for $k$-subset sampling [1, 2, 3]. As mentioned above, the connection drawn to Gumbel-Softmax is merely an interesting aside; our main focus throughout the paper has been with **subsets of $k$ elements**. We do, however, compare against Gumbel-Softmax on the DVAE experiment proposed by the authors of the Gumbel-Softmax gradient estimator.
>
> [Improper comparison with Gumbel Softmax estimator]
>
> As already pointed out, our main focus in the paper is sampling **subets** of elements, rather than a single element. Therefore, most of our experiments are concerned with the former, rather than the latter; after all the former is merely an edge case of our problem statement.
>
> Nonetheless, we do compare against the Straight-Through Gumbel Softmax estimator, both qualitatively, i.e. bias and variance, (c.f. Section 4), and quantitatively on the task of generative learning of MNIST digits using Discrete VAEs, as in the original Gumbel Softmax paper (c.f. Section 5.2 and figure 4 (right) in our paper, and Section 4.2 in [4]).
>
> We would like to emphasize that there is a mismatch in notation between [4] and our paper: In our paper, we use $k$ in reference to the size of the **subset** of elements to be sampled, where as [4] use $k$ to denote **the number of classes in the categorical distribution**. Consequently, in our paper, we use $k=1$ to denote the case where we’re interested in sampling a subset of size $1$, or in other words, a single element from a categorical fashion much like is done in [4]. In fact, in section 4, the *bias and variance are computed using 10000 samples, not only 1*. *We reiterate, the statement  “choose number of samples for it greater than 1 as authors did to match running time of their estimator.” is incorrect*.
>
> Regarding runtime, we would like to point out that, for $k=1$, SIMPLE’s runtime no longer depends on $k$. In fact, the runtime matches Gumbel Softmax, when $k=1$. To see this, please consider the below implementations, in PyTorch, of Gumbel-Softmax and SIMPLE, respectively.
>
>
>     def sample_gumbel(shape, eps=1e-20):
>         U = torch.rand(shape)
>         return -Variable(torch.log(-torch.log(U + eps) + eps))
>
>     def gumbel_softmax_sample(logits, temperature):
>         y = logits + sample_gumbel(logits.size())
>         return F.softmax(y / temperature, dim=-1)
>
>     def gumbel_softmax(logits, temperature=1.0):
>         y = gumbel_softmax_sample(logits, temperature)
>        shape = y.size()
>         _, ind = y.max(dim=-1)
>         y_hard = torch.zeros_like(y).view(-1, shape[-1])
>         y_hard.scatter_(1, ind.view(-1, 1), 1)
>         y_hard = y_hard.view(*shape)
>         return (y_hard - y).detach() + y
>
>     def SIMPLE(logits):
>         y = F.softmax(logits, dim=-1)
>         y_perturbed = F.softmax(logits + sample_gumbel(logits.size()))
>         shape = y.size()
>         _, ind = y_perturbed.max(dim=-1)
>         y_hard = torch.zeros_like(y_perturbed).view(-1, shape[-1])
>         y_hard.scatter_(1, ind.view(-1, 1), 1)
>         y_hard = y_hard.view(*shape)
>         return (y_hard - y).detach() + y

---

> > ### Author Response · Authors · 2022-11-10
> > **Response to Reviewer pM21 continued**
> >
> > [Why do GS and proposed estimator have bias]
> >
> > Gumbel-Softmax is not based on the log-derivative trick, that would be the Score function estimator (SFE), which we compare against in Figure 1, and which is indeed unbiased, which is corroborated by Figure 1. Rather, Gumbel Softmax uses the straight-through estimator to reparameterize the gradient of the loss function w.r.t the sample as a gradient of the loss function w.r.t the perturbed marginals (the output of gumbel_softmax_sample) above, and considers the gradient of the samples w.r.t the marginals to be the identity (see Section 2.2 in [4]). The same trick is utilized here, in our paper to derive SIMPLE. So, while we do obtain exact samples on the forward pass, we do need a means of bypassing the non-differentiability of discrete sampling, and thus our use of the straight-through estimator.
> >
> > We hope this clears up any confusion and concerns, and if so, we’re hoping to turn the verdict into an accept.
> >
> > References:
> >
> > [1] Jianbo Chen, Le Song, Martin Wainwright, and Michael Jordan. Learning to explain: An information- theoretic perspective on model interpretation. In Proceed- ings of the 35th International Conference on Machine Learning.
> >
> > [2] Sang Michael Xie and Stefano Ermon. Reparameterizable subset sampling via continuous relaxations. In Proceedings of the Twenty-Eighth International Joint Conference on Artificial Intelligence.
> >
> > [3] Mathias Niepert, Pasquale Minervini, and Luca Franceschi. Implicit mle: Backpropagating through discrete exponential family distributions. Advances in Neural Information Processing Systems, 34, 2021.
> >
> > [4] Eric Jang, Shixiang Gu, and Ben Poole. Categorical reparameterization with gumbel-softmax. 5th International Conference on Learning Representations.

---

> > > ### Comment · Reviewer_pM21 · 2022-11-15
> > > **Thanks**
> > >
> > > Thank you for clarifications, I am convinced and raise my score

---

### Official Review · Reviewer_XcRS · 2022-10-24

**Confidence:** 4
**Correctness:** 2
**Technical Novelty And Significance:** 3
**Empirical Novelty And Significance:** 2
**Recommendation:** 6

**Clarity, Quality, Novelty And Reproducibility:**

This paper is reasonably clear.  The contributions appear to be novel, with good empirical and theoretical contributions.  Although the authors do not currently provide source code, enough detail appears to be present in the paper to allow the proposed algorithms to be reproduced.

**Strength And Weaknesses:**

Strengths:
* Compared to competing approaches for computing gradients for k-subset sampling, SIMPLE provides significant reductions in bias and variance.  Compared to competing approaches, SIMPLE can also provide moderate to significant improvements in predictive performance when used for training models, as shown by the experimental results in this paper for the learning to explain task.
* When used to compute the ELBO for a $k$-subset discrete VAE, compared to competing approaches SIMPLE can provide significantly lower ELBO loss on the test set.
* The complexity analysis presented in this paper shows that the proposed SIMPLE algorithm appears to scale when computing the required marginal probabilities and sampling k-subsets, particularly when using vectorized computations for parallelization.

Weaknesses:
* The complexity analysis provided in the paper focuses on algorithms for computing marginals probabilities and for sampling $k$-subsets.  However, an explicit analysis of the complexity for computing the backward pass (from Algorithm 3) appears to be missing.  This should be provided in the paper.
* This paper does not include a discussion of the weaknesses/drawbacks of the proposed approach.  For example, SIMPLE seems to assume that the $k$-subset distribution can be factorized as a product of the conditional marginals; that is, the subset is a multilinear function of its elements (as pointed out in the first paragraph of Sec. 3.1).  However, this assumption may only be valid in cases where the $k$-subset distribution does not involve higher-order interactions between elements within a subset, such as subsets that are modeled using a Transformer-based model.  An explicit discussion of this limitation should be provided in the paper.
* An empirical evaluation of the runtime of SIMPLE, compared to the competing approaches, is missing and should be provided.  While the complexity analysis appears to show that SIMPLE scales, an empirical validation of this scalability is important.



**Summary Of The Paper:**

This paper presents an approach for gradient estimation for discrete k-subset sampling, called SIMPLE.  For the forward pass, this approach involves discrete sampling.  For the backward pass, SIMPLE efficiently computes the gradient with respect to the exact marginals of the $k$-subset distribution.  Compared to competing approaches for gradient estimation in this setting, SIMPLE exhibits lower bias and variance.  In addition to an efficient algorithm for computing the gradient, the authors also present an algorithm for computing the exact ELBO for $k$-subset distribution, leading to state-of-the-art performance on sparse VAE learning.

**Summary Of The Review:**

There are some important empirical and theoretical contributions in this paper. The experimental results are somewhat convincing, showing that the proposed SIMPLE method outperforms a number of competing methods for estimating gradients for the discrete k-subset setting, in terms of both reduced bias and variance, and improved predictive performance.  However, there are several issues with this paper that should be addressed by the authors, as mentioned in the list of weaknesses above.  Overall, in terms of acceptance, this is a borderline paper in its current state.

---

> ### Author Response · Authors · 2022-11-10
> **Response to Reviewer XcRS**
>
> We would like to thank the reviewer for their feedback.
>
> [complexity analysis]
>
> The complexity of the backward pass as shown in Algorithm 3 is a constant multiple of the time to evaluate the log marginals, $\log a[n,k]$ since it calls auto-diff (https://mblondel.org/teaching/autodiff-2020.pdf) and the complexity analysis of computing the marginals is presented in Sec. 6. We will append this discussion to Sec. 6 for a full analysis of the algorithm complexity.
>
>
> [factorized distribution assumption]
>
> The factorized distribution assumption is general where each discrete random variable has a category probability possibly parameterized by the neural network. This assumption is shared among all the previous work on learning k-subset distributions including I-MLE (Niepert et al., 2021), SoftSub (Xie & Ermon, 2019) and L2X (Chen et al. 2018).
>
> [empirical runtime]
>
> We provide the runtime of both SIMPLE and SoftSub for synthetic experiments: SIMPLE 3.40 ms ± 6.08 µs per loop and SoftSub 1.11 ms ± 13 µs per loop, as an average over seven runs. Their similar runtimes are consistent with our complexity analysis that both of them have non-vectorized complexity $O(nk)$. For vectorized complexity, we’ve shown that our algorithm can be optimized to achieve complexity $O(log k \log n)$ assuming perfect parallelization; while the vectorized complexity of SoftSub can be optimized to at most $O(k \log n)$ where the factor $k$ is due to their use of the relaxed top-$k$ operation.
>
> We hope this clears up any confusion and concerns, and if so, we’re hoping to turn the verdict into an accept.

---

> > ### Comment · Reviewer_XcRS · 2022-11-15
> > **Rebuttal response**
> >
> > I thank the authors for their rebuttal comments and explanations.  I have read all reviews and the rebuttal comments from the authors.  The authors have mostly addressed the novelty concerns raised by some reviewers.  However, some gaps in this paper remain.   Specifically, I would like to see a more complete analysis of the empirical runtime of SIMPLE compared to competing approaches, for various values of $n$ and $k$, to empirically validate the complexity analysis provided in Sec. 6.  Also, I would still like to see a discussion of the weaknesses/drawbacks of the proposed SIMPLE approach, including the limitations of the assumption that the $k$-subset distribution can be factorized as a product of the conditional marginals (even if this assumption is shared by prior work).  Therefore, my score of 6 (marginally above the acceptance threshold) remains unchanged.

---

> > > ### Author Response · Authors · 2022-11-21
> > > **Response to Reviewer XcRS**
> > >
> > > Thanks for engaging with us, and apologies for the late reply.
> > >
> > > Below is a comparison of the the run times of SoftSub (first entry in every cell) and SIMPLE (second entry in every cell, after the comma). All times are in milliseconds.
> > >
> > > |      |     k=5    |    k=10    |    k=15    |
> > > |:----:|:----------:|:----------:|:----------:|
> > > | n=10 | 5.27, 7.21 |      /     |      /     |
> > > | n=15 |  5.35, 9.9 | 7.85, 8.35 |      /     |
> > > | n=20 | 5.27, 10.4 | 9.14, 10.5 | 12.6, 10.6 |
> > >
> > > And here are some further data points:
> > >
> > > For $n=40$ and  $k =25$:  20.1, 23.8
> > >
> > > For $n=80$, $k=50$:  32.7, 43.0
> > >
> > > For $n=100$, $k=80$: 39.5, 53.5
> > >
> > > [Factorized-distribution assumption]:
> > >
> > > In this paper, similar to all previous works on $k$-subset selection, we assume that the items are picked independently. We agree that relaxing the factorized distribution assumption, which assumes the absence of correlations between items included in the subset, beyond the fact that there’s $k$ of them, could be beneficial in some cases. For instance, one might expect that in the L2X task, the presence of a word such as “delicious” might correlate strongly with the presence of a word such as “good”. In the presence of such correlations, however, the $k$-subset distribution becomes intractable. We do think that this is an interesting problem, but a non-trivial one, nonetheless, that we look forward to addressing in our future work.
> > >
> > > We will add the following line to our related works section: “All of the above works, ours included, assume the independence of the selected items, beyond there being $k$ of them. That is with the exception of [1] which make use of a relaxation using pairwise embeddings, but do not make their code available. We view extending our work to distributions where the $k$ items are correlated as a very interesting future endeavor.
> > >
> > > We hope this lays to rest any concerns you might have regarding our paper.
> > >
> > > References:
> > >
> > > [1] Max B. Paulus, Dami Choi, Daniel Tarlow, Andreas Krause, and Chris J. Maddison. 2020. Gradient estimation with stochastic softmax tricks. In Proceedings of the 34th International Conference on Neural Information Processing Systems."

---

### Author Response · Authors · 2022-11-10
**General Response**

Based on the feedback from reviewers, we’ve uploaded a revised version of the submission with improved clarity as well as the code of all experiments for reproducibility.

We would also like to address the novelty concerns raised by some reviewers by reiterating the significance of the problem at hand, as well as our contributions. In a nutshell, the problem of differentiable k-subset sampling is an integral part of machine learning [1,2,3]. One important attempt at gradient estimation for k-subset sampling, SoftSub, generalizes Gumbel-Softmax, returning relaxed samples, but biasing the gradient estimate in the process. Moreover, relaxed samples cannot be used in settings such as discrete VAEs. The key idea we identify is, there is no need to relax the samples, but instead, we can discretely sample the $k$-subset distribution, and estimate the gradient as the derivative of the marginals of the $k$-subset distribution, all done tractably. This is a significant and novel contribution, conceptually, and practically, as shown in the experiments.

References:

[1] Jianbo Chen, Le Song, Martin Wainwright, and Michael Jordan. Learning to explain: An information- theoretic perspective on model interpretation. In Proceed- ings of the 35th International Conference on Machine Learning.

[2] Sang Michael Xie and Stefano Ermon. Reparameterizable subset sampling via continuous relaxations. In Proceedings of the Twenty-Eighth International Joint Conference on Artificial Intelligence.

[3] Mathias Niepert, Pasquale Minervini, and Luca Franceschi. Implicit mle: Backpropagating through discrete exponential family distributions. Advances in Neural Information Processing Systems, 34, 2021.

---

### Decision · Program_Chairs · 2023-01-20

**Decision:**

Accept: poster

**Justification For Why Not Higher Score:**

Good paper overall: interesting idea, well-written, nice experiments, mostly cites the literature properly.

**Justification For Why Not Lower Score:**

A bit incremental, could include more baselines in the comparisons.

**Metareview: Summary, Strengths And Weaknesses:**

The paper proposes a novel gradient estimator for k-subset sampling.

Strengths:
- A new gradient estimator for k-subset sampling
- Well-written paper
- Several illustrative experiments

Weaknesses:
- Comparisons with other methods could be more thorough

A reviewer was negative about the clarity of the paper but I feel the authors have addressed the reviewer's concerns. Unfortunately, the reviewer didn't update their score. Hence I think the average score of 5.25 does not give justice to this paper.

I recommend acceptance as poster.

Papers possibly worth mentioning:
- https://proceedings.mlr.press/v97/kool19a.html
- https://arxiv.org/abs/1910.11369 (knapsack polytope section)

**Note From Pc:**

if the above contains the word "oral" or "spotlight" please see: "oral" presentation means -> notable-top-5% and "spotlight" means -> notable-top-25%. As stated in our emails, we are disassociating presentation type from AC recommendations